# Multistate animal-contact-related nontyphoidal *Salmonella enterica* outbreaks in the United States, 2009–2022: Network and machine learning analyses of exposure sources, settings, and serovars

**Hammad Ur Rehman Bajwa[1], Suman Bhowmick[1], Csaba Varga[1,2]***

**1** University of Illinois, College of Veterinary Medicine, Urbana-Champaign, Illinois, United States of America, **2** University of Illinois, Carl R. Woese Institute for Genomic Biology, Urbana-Champaign, Illinois, United States of America

* cvarga@illinois.edu

## Abstract

### Background

Nontyphoidal *Salmonella enterica* (NTS) is a major public-health threat in the United States of America (U.S.). Evaluating associations between serovars, exposure sources, and settings in multistate outbreaks can reveal the drivers of NTS transmission and guide prioritization of targeted prevention and control strategies.

### Methods

We analyzed multistate animal-contact-related NTS outbreaks reported to the CDC National Outbreak Reporting System during 2009–2022. We calculated incidence rates (IR) per 10 million population-years (MPY) and assessed temporal trends in IRs using Joinpoint regression. We constructed interstate co-occurrence networks linking serovars, exposure sources, settings, and states, and applied a random forest classifier to identify variables most useful for distinguishing outbreak profiles.

### Results

We identified 177 multistate outbreaks (0.06 per 10 MPY) involving 40 serovars. Incidence significantly declined from 2009 to 2013 and remained stable thereafter. Random forest rankings identified birds and reptiles as the most influential exposure sources and agricultural feed stores and residential homes as the most influential exposure settings in distinguishing outbreak profiles. Co-occurrence network analysis revealed two major communities. The first included outbreaks involving serovars Enteritidis and Infantis, bird exposure source, and agricultural feed stores or farms as exposure settings, with co-occurrence hubs across the Midwest, Northeast, and

**Data availability statement:** Publicly available data used in this study are accessible through the CDC BEAM Dashboard (https://www.cdc.gov/beam/dashboard/index.html). This study also analyzed more detailed, non-public data obtained under a data use agreement. These restricted data are not publicly available, and the authors did not have special access privileges beyond the terms of the agreement. Third-party data requests should be directed to NORS staff at NORSAdmin@cdc.gov.

**Funding:** The author(s) received no specific funding for this work.

**Competing interests:** The authors have declared that no competing interests exist.

Southern regions. The second community involved outbreaks linked with reptiles and mammals as exposure sources, residential homes and farms as exposure settings, and serovars Hadar, Typhimurium, and Braenderup, which were co-occurring in the Western and Southern regions.

## Conclusions

Multistate animal-contact NTS outbreaks clustered into distinct serovar-exposure, source, setting, and region patterns, suggesting different NTS outbreak transmission pathways. The persistence of NTS serovars across states, diverse animal-contact sources, and exposure settings underscores the ongoing zoonotic transmission risk at the human-animal and environmental interfaces. A region-specific One Health approach to prevent and control NTS outbreaks is suggested to reduce the health burden.

## Introduction

Nontyphoidal *Salmonella enterica* (NTS) is an important zoonotic pathogen and a leading global public-health concern [1]. Globally, NTS causes over 95 million enteric illnesses, with more than 500,000 invasive infections, leading to high fatality rates [2]. It also poses an economic burden, contributing to millions of disability-adjusted life years and to substantial healthcare costs worldwide [3]. Most NTS infections are self-limiting; however, severe invasive disease can occur [4]. The NTS bacteria are adaptable to various hosts, with NTS serovars such as Typhimurium, Enteritidis, Heidelberg, and Infantis being common causes of human enteric disease, including in the U.S. [5]. Worldwide, NTS causes millions of illnesses annually, with young children, older adults, and immunocompromised persons at elevated risk of severe disease [6]. Although foodborne transmission accounts for a large proportion of NTS infections, illnesses associated with direct animal contact also contribute to the disease burden [7,8]. Humans can be infected by contact with infected animals or their contaminated environment at farms, live-animal markets, petting zoos, and households with companion or exotic animals, requiring integrated One Health prevention strategies [9].

Outbreaks linked to animal contact are prone to cross-jurisdictional spread as modern animal production and distribution networks regularly move live animals across state lines, so single contamination points, such as NTS-infected day-old chicks from a hatchery supplying multiple states, a common breeder distributing reptiles or other pet animals to pet stores and private owners, can cause geographically dispersed outbreaks [10]. These distribution-driven pathways enable multistate dissemination and need a coordinated national surveillance, interstate response, and targeted control measures to identify sources and interrupt transmission [11]. Multistate NTS outbreaks linked to poultry, reptiles, and mammalian species are characterized by prolonged exposure windows and diffuse transmission pathways, which complicate case finding, source attribution, and timely intervention [11,12]. Although these animal contact-related events do not represent the majority of reported outbreaks, surveillance analyses show they account for a disproportionate share of

illnesses, hospitalizations, and deaths, underscoring their public-health impact and the need for strengthened, coordinated surveillance and control efforts [13].

In the US, the NTS multistate animal contact-related outbreaks have been linked to specific animal contact sources, including poultry [14], reptiles (turtles and bearded dragons) [15], and exotic pets [16]. The recurrent multistate outbreaks that are associated with live poultry demonstrate how lapses in biosecurity in mail-order hatcheries, shipping practices, and farm distribution systems enable the dissemination of pathogens across the country, and call for upstream interventions and strengthened trace-back capacity [11,12].

Reptile-associated multistate NTS outbreaks have persisted over the past decade, driven in part by illegal trade and sale of prohibited reptile species despite federal restrictions. Ongoing illicit distribution, via informal channels, unregulated online marketplaces, and noncompliant retailers, facilitates cross-state dissemination of infected animals, complicates trace-back investigations, and sustains transmission, highlighting the need for strengthened enforcement, targeted public education, and upstream interventions at points of supply [16]. Although less frequent than poultry or reptile-associated outbreaks, multistate NTS outbreaks have been associated with exotic pets and wild birds [16,17]. The diversity of animal reservoirs for NTS complicates surveillance, source attribution, and control; however, it creates intervention opportunities for public-health authorities [18].

Furthermore, animal contact-related NTS outbreaks are not only characterized by exposure pathways but also by serovar heterogeneity [13]. There are serovars such as Typhimurium [16], Enteritidis [17], and Infantis [19] that are frequently reported in single- and multistate NTS outbreaks.

Despite recognized NTS diversity, most multistate outbreak surveillance studies focus on individual serovars, sources, and settings, and do not explain patterns across common and rare serovars. Studies that have focused on genetic evaluation of NTS serovars and they have reported that understanding the transmission dynamics of outbreaks and their source attribution is important [19,20]. No previous study explored long-term, NTS serovar-level animal contact-related multistate outbreaks in the US, and the relationship between serovars and animal contact sources remains unclear. Previous epidemiological studies focused on individual serovars or specific exposure sources, limiting the full scope of NTS transmission dynamics across the U.S. [16,21,22]. In multistate outbreaks, interconnected events might involve shared distribution systems, supply chains, and occupational exposures [21], and an integrated analytical approach is needed to quantify NTS burden and identify factors that sustain its endemicity across various reservoirs and regions.

We hypothesized that the epidemiology of animal-contact-related multistate NTS outbreaks varies by serovar, exposure source, and setting, and is influenced by interstate transmission pathways. To test this hypothesis, we: 1) quantified the burden of animal-contact–associated multistate NTS outbreaks in the United States (2009–2022); 2) assessed temporal changes in outbreak incidence rates using trend analyses; 3) evaluated the relative importance of geographic location, exposure source, and setting in distinguishing outbreak profiles using a random forest classifier; and 4) mapped and analyzed co-occurrence patterns among serovars, states, exposure sources, and settings using network analysis.

## Materials and methods

### Data source, study design, and setting

The National Outbreak Reporting System (NORS) is a voluntary disease surveillance system managed by the Centers for Disease Control and Prevention (CDC) that collects reports on enteric disease outbreaks, including those transmitted through animal contact [23,24]. The system is used by local health departments, state-level health authorities, and the CDC to report and track outbreaks [24]. An outbreak is defined as the occurrence of more cases than expected within a period, a specific location, and a target population [25]. Multistate (more than 2 states) NTS outbreak investigations are coordinated and reported to NORS by CDC staff, while single-state outbreaks are investigated and reported to NORS by single jurisdictions [26].

The animal contact-related NTS outbreak data were obtained from the CDC through a dataset request [27]. The dataset included all animal-contact-related NTS outbreaks reported in NORS from 2009 to 2022. The data included details on

NTS outbreak occurrence dates, locations (States), animal contact exposure sources, and settings, available in NORS as of August 2024. This retrospective observational study analyzes human NTS animal contact-related multistate outbreaks. It includes all outbreaks with confirmed or suspected status, along with data on serovars, exposure states, animal contact exposure sources, and settings. Outbreaks with "not available" or "other" categories for serovars, exposure sources, and settings were treated as distinct categories for further analysis.

Population data for the study period were obtained from the U.S. Census Bureau [28]. The state administrative boundary shapefile was obtained from the Topologically Integrated Geographic Encoding and Referencing (TIGER) /Line database from the U.S. Census Bureau [29].

## Data management

The dataset was assessed for completeness, consistency, and accuracy. Each outbreak had a unique ID (CDCID), which was used to detect duplicate or incomplete records. In multistate outbreaks, each outbreak has a distinct CDCID, and that is shared across all participating states. The CDCID was used to verify outbreak grouping and aggregation of multistate outbreaks. The raw dataset consisted of multiple spreadsheets, each representing specific information (i.e., mode of transmission, etiology, geographical information, case counts, exposure sources, and settings). The spreadsheets were processed and merged based on their CDCID using R packages, including readxl [30], writexl [31], dp1yr [32], and tidyr [33]. The animal contact exposure sources were categorized using the CDC's categorization scheme to harmonize exposures [34]. The categorical variables, including serovars, states, exposure sources, and settings, were standardized and coded in structured formats, including binary matrices.

## Assessing NTS outbreak incidence rates

The R software (Version 4.5.2) in the RStudio Platform (Version 2026.04.0; 2009–2026 Posit Software, PBC) was used for all statistical analyses [35].

The national multistate outbreak incidence rates (IRs) per 10-million person-years (10 MPY) for 2009–2022 were calculated, as

$$IR = \frac{N}{P_*} \text{x } 10,000,000$$

N = Total number of multistate outbreaks in a year
P* = Sum of populations of all states involved (N times)
N = number of times the state appeared in the multistate outbreak in that year.

Table 1 summarizes the explanatory variables (features) used in the analysis, including their description, type, category, and the specific analysis in which they were applied.

## Temporal trend analysis using the Joinpoint regression method

Using the Joinpoint regression (JPR) analysis software (Version 5.4.0.0, 2025, Surveillance Research Program, National Cancer Institute, Maryland, USA), national trends in IRs of NTS multistate outbreaks across the US from 2009 to 2022 were examined [36]. Joinpoint analysis is a statistical approach used to identify points (joinpoints) where the trend in a dataset changes direction [36]. The temporal trends were assessed with Annual Percent Change (APC) to quantify the average yearly percent change within each Joinpoint-defined segment. APC was calculated from log-linear regression models as:

$$APC = (e^{\beta} - 1) \times 100$$

**Table 1. Summary of variables used for the analysis of Nontyphoidal *Salmonella enterica* animal contact-related multistate outbreaks across the US, 2009–2022.**

| Variable | Description | Type | Category | Analysis |
|---|---|---|---|---|
| Year | Outbreak occurrence time | Numeric | 2009–2022 | Outbreak IRs; Joinpoint Regression Analysis, |
| States | Geographical locations of outbreak occurrence | Categorical | 48 states and the District of Colombia | Predictors: Random Forest and Network Analysis |
| State Indicators | Binary representatives of states | Binary | 0 = absent 1 = present | Predictors: Random Forest and Network Analysis |
| Serovars | *Salmonella enterica* serovars | Categorical | 40 serovars | Outbreak IRs and Network Analysis |
| Serovars Indicators | Binary representatives of serovars | Binary | 0 = absent 1 = present | Network Analysis |
| Animal contact exposure sources | Animal contact reservoirs (e.g., birds, mammals, and reptiles) | Categorical | Animal contact exposure source categories | Random Forest and Network Analysis |
| Animal contact exposure source indicators | Binary representatives of exposure sources | Binary | 0 = absent 1 = present | Random Forest and Network Analysis |
| Animal contact exposure settings | Animal contact exposure sources (e.g., feed stores, educational settings, farm/dairy/agricultural settings) | Categorical | Animal contact exposure settings | Random Forest and Network Analysis |
| Animal contact exposure settings indicators | Binary representatives of exposure settings | Binary | 0 = absent 1 = present | Random Forest and Network Analysis |
| Outbreak Incidence rates (IRs) | Annual/Mean Outbreak IRs | Continuous | | Outcome: Joinpoint Regression Analysis |
| Co-occurrence network (edges) | Co-occurrence among variables in a multistate outbreak | Weighted | Frequency of Occurrence (size) | Network Analysis |
| Co-occurrence network (Nodes) | Serovars, states, exposure sources, and settings | Categorical | All variables | Network Analysis |

Here β represents the slope of the fitted model. The average annual percentage change (AAPC) was used to summarize overall trends across the full study period.

The results of the JPR models were illustrated in graphs.

## Random Forest analysis of multi-state outbreak data

We analyzed the binary dataset of human NTS multistate outbreaks from 2009 to 2022 using the Random Forest (RF) modeling approach. The RF method is a flexible, non-parametric ensemble learning technique that can perform well with high-dimensional datasets, which contain correlated and binary predictors, as our surveillance dataset carries the same characteristics. The RF method can also model complex, non-linear relationships among the higher-dimensional model variables, and it can also include higher-order interactions without requiring explicit specifications. In addition, RF also incorporates internally validated performance estimates through out-of-bag (OOB) error and can generate interpretable variable importance measures, which were central to our objective of identifying the key driving factors distinguishing cluster membership into the model [37]. The RF model is less sensitive to multicollinearity and overfitting than traditional regression-based approaches, and is suitable for exploratory analyses of complex multistate outbreak data with many potentially correlated indicators. Moreover, we outline our classification methodology and explore the relationships among key predictors. The RF classifier [38] was applied to a binary predictor matrix capturing features of multistate NTS outbreaks, including state indicators, exposure sources, and exposure settings. We removed variables with no variability before our analysis, and no additional feature selection or dimensionality reduction techniques were applied before RF

model fitting. We performed unsupervised hierarchical clustering on the binary matrix using Jaccard dissimilarity and Ward's linkage, and then considered the resulting dendrogram, which was cut into four clusters [39]. We treated the cluster membership as a 4-level categorical outcome in an RF model [40]. The outcome variable in the RF model was the cluster membership of multistate NTS outbreaks, categorized into four distinct groups based on similarities in their predictors (Table 1). In our model exploration, we included all remaining surveillance variables as predictors, and the variable importance was summarized using mean decrease in accuracy and mean decrease in Gini coefficient. Collectively, the above-mentioned steps involved identifying clusters through unsupervised methods, followed by the application of an RF classifier to determine the variables that distinguish cluster membership of the surveillance data (Table 1).

Additionally, RF constructs multiple decision trees using bootstrap samples of the data and random subsets of predictors at each split, with final classification determined by majority vote across trees [38]. Model performance and predictor relevance were evaluated using standard random-forest importance metrics: mean decrease in Gini (reflecting each variable's contribution to node purity across trees) and permutation-based mean decrease in accuracy (quantifying the loss in classification accuracy after the random permutation of each predictor). We conducted all analyses in R using the following packages: readxl [30], tidyverse [41], dplyr [32], tidyr [33], randomForest [38], vegan [42], uwot [43], ggplot2 [44], and pheatmap [45].

### Network analysis of multi-state outbreak data

The co-occurrence network approach was used to characterize relationships (co-occurrences) among NTS serovars, animal-contact exposure sources, settings, and States in multistate outbreaks [46]. This approach enables the identification of co-occurrence patterns and clustering structures across outbreak features. In our study, an undirected network was developed because the outbreak surveillance data showed shared participation in the same outbreak, and no directional transmission pathways could be established. To display the frequency of co-occurrence between categorical variables, a weighted network structure was used to allow for more recurrent relationships within the network [47].

We constructed a weighted, undirected outbreak co-occurrence network [48] where nodes represented U.S. states, exposure sources (e.g., birds, reptiles), exposure settings (e.g., agricultural feed stores, daycare centers), and serovars (Table 1). Edges represented shared participation in the same multistate outbreak, with edge weights corresponding to the number of outbreaks involving each pair of nodes. For the network analysis, only the 48 contiguous U.S. states plus the District of Columbia were included due to spatial analysis requirements, as the inclusion of Alaska and Hawaii would have introduced geographic distortions in the spatial relationships between states. The U.S. state boundary shapefiles were integrated as polygon layers to provide geographic context for visualizing the spatial co-occurrence network. We further described this network as weighted, where edge weights represented the magnitude of co-occurrence between node pairs, thus allowing stronger and weaker associations to be distinguished and visually represented. Network metrics, including degree, betweenness, clustering coefficient, and closeness, were calculated to characterize state-level connectivity. Louvain community detection was applied to identify clusters of related states, serovars, exposure sources, and settings. The R packages that were used to explore co-occurrence network patterns were igraph [46], ggraph [49], ggplot2 [44], and networkD3 [48].

## Results

### Assessing NTS outbreak incidence rates

A total of 177 NTS multistate outbreaks were reported in the US from 2009 to 2022, with a mean outbreak IR of 0.06 per 10 MPY. The highest outbreak IR was reported in 2009 (0.10 per 10 MPY) and 2018 (0.08 per 10MPY), and the lowest in 2013 (0.05 per 10 MPY). The highest number of states (52) involved in multistate outbreaks occurred in 2022, while the lowest (44) occurred in 2010. Additionally, 2022 also saw the highest number of state mentions (341) (S1 Table).

## Temporal trend analysis using the Joinpoint regression method

The Joinpoint regression analysis was performed at the national level to explore NTS temporal trends from 2009 to 2022 (Fig 1 and S2 Table). The Joinpoint Regression analysis presents the Annual Percent Change in NTS outbreak trends from 2009 to 2022. The final selected model with a single Joinpoint in 2013, distinguished two trend periods, a steep and significant decline from 2009 to 2013, with an APC of −10.4%, 95% CI: −23.6–4.2, and p-value = 0.001, and a non-significant decreasing trend in later years from 2013 to 2022 (APC = −0.3%, 95% CI: −2.4–6.8, p-value = 0.98). (Fig 1 and S2 Table).

## Random Forest analysis of NTS outbreak data

Within the Random Forest framework, the U.S. States with the highest importance scores, indicating their strong and consistent contributions to distinguishing the outcome variable (cluster membership of multistate NTS outbreaks, categorized into four distinct groups based on similarities in their predictors), included Ohio, Indiana, Nebraska, Vermont, Pennsylvania, Idaho, Michigan, Texas, Kentucky, Alabama, Illinois, and Minnesota (**Fig 2** and S3 Table). These states, with the highest importance scores, were more likely to be represented in the four multistate NTS outbreak clusters, as their characteristics significantly influenced cluster membership (**Fig 2**).

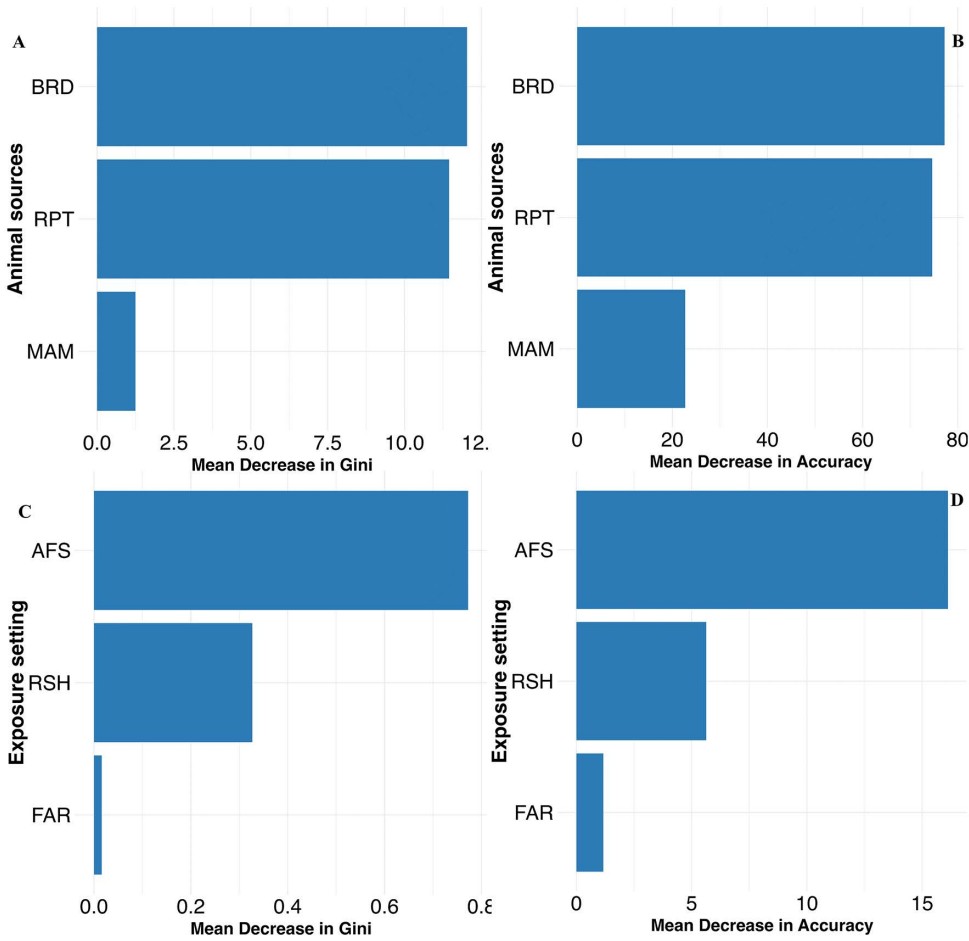

**Fig 1. Joinpoint regression analysis of NTS animal contact-related multistate outbreaks across the US from 2009-2022.**

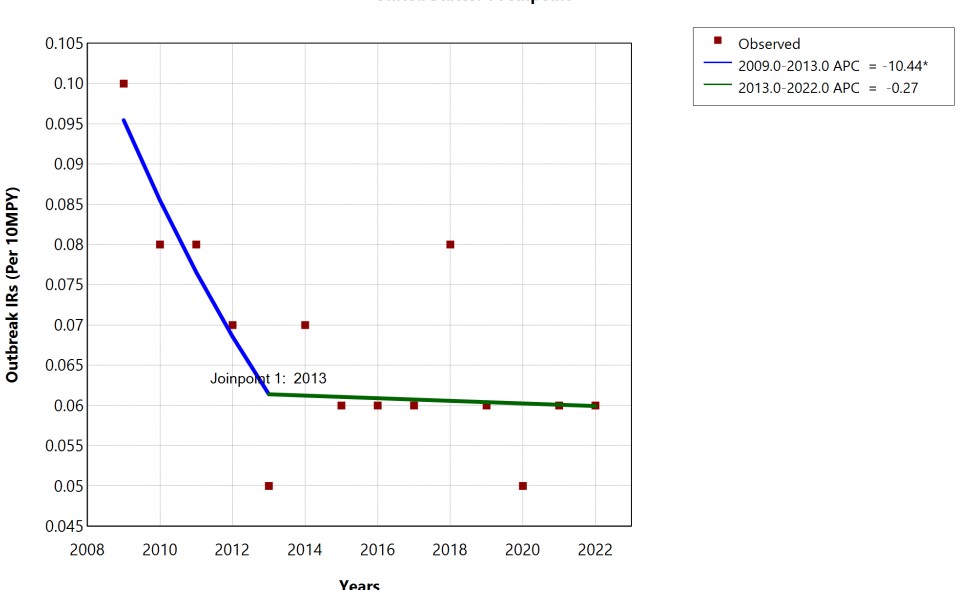

United States: 1 Joinpoint

Legend:
- Observed
- 2009.0-2013.0 APC = -10.44*
- 2013.0-2022.0 APC = -0.27

Joinpoint 1: 2013

X-axis: Years
Y-axis: Outbreak IRs (Per 10MPY)

* Indicates that the Annual Percent Change (APC) is significantly different from zero at the alpha = 0.05 level.
Final Selected Model: 1 Joinpoint.

**Fig 2. State-level variable importance for nontyphoidal *Salmonella enterica* animal-contact multistate outbreaks from random forest classification. A.** Mean decrease in Gini impurity; **B.** Mean decrease in accuracy. The plots show the relative contribution of each U.S. state to the classifier's ability to distinguish outbreak profiles, with higher values indicating greater importance. States (abbreviations): NE-Nebraska; OH-Ohio; IN-Indiana; MN-Minnesota; VT-Vermont; ID-Idaho; IL-Illinois; OR-Oregon; MI-Michigan; MO-Missouri; AL-Alabama; UT-Utah; PA-Pennsylvania; KY-Kentucky; CO-Colorado; OK-Oklahoma; TX-Texas; VA-Virginia; MA-Massachusetts; WI-Wisconsin; GA-Georgia; MD-Maryland; MT-Montana; WA-Washington; CT-Connecticut; TN-Tennessee; ME-Maine; LA-Louisiana; KS-Kansas; AZ-Arizona; AR-Arkansas; FL-Florida; NV-Nevada; MS-Mississippi; DE-Delaware; HI-Hawaii; AK-Alaska.

Birds and reptiles as the most influential animal-contact exposure sources, and agricultural feed stores and residential homes as the most influential exposure settings, were identified by the Random Forest model, based on variable importance ranking, suggesting that they all played an important role in defining the four multistate NTS outbreak clusters (outcome variable) (**Fig 3** and S3 Table).

## Network analysis of NTS outbreak data

A spatially explicit co-occurrence network was constructed for multistate outbreaks, integrating U.S. states, NTS serovars, animal exposure sources (Birds, Mammals, Reptiles, Amphibians, Fish, NA, Others), and settings (Agricultural feed stores, farms, residential settings, daycare centers, educational institutions, and others) (**Fig 4**). The centroids of U.S. states were used as one of the nodes and also as the spatial reference to aid in visualizing patterns of co-occurrence across different outbreak characteristics. Edges represented shared participation of nodes in the same outbreak (co-occurrence), with edge weights equal to the number of outbreaks in which each pair of nodes contributed. Several network metrics were calculated, including degree, which is the number of connections each node has, strength which is the frequency of shared outbreaks, betweenness, which represents the number of times a node is on the shortest path between two nodes, closeness, which determines how close the node is to all other nodes in the network, clustering coefficient, which is a measurement of the local connectivity, and Louvain communities, which represents a group of densely connected nodes.

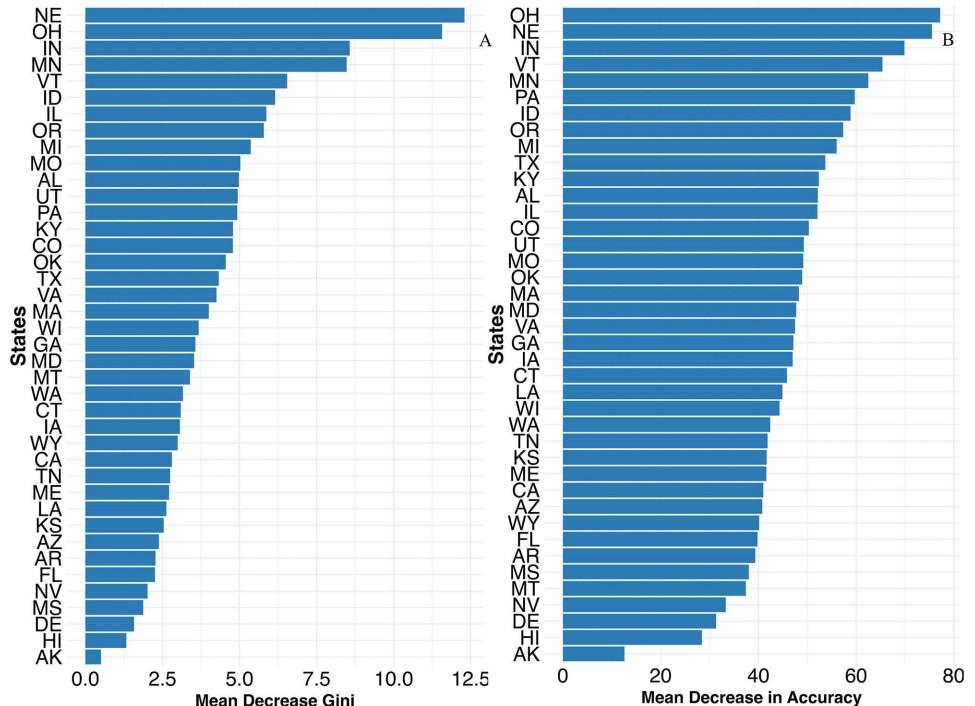

**Fig 3. Variable importance of exposure sources and settings in the random forest model for nontyphoidal *Salmonella enterica* animal-contact multistate outbreaks. A.** Exposure sources – Mean decrease in Gini; **B.** Exposure sources – Mean decrease in accuracy; **C.** Exposure settings – Mean decrease in Gini; **D.** Exposure settings – Mean decrease in accuracy. Abbreviations: BRD = Birds; MAM = Mammals; RPT = Reptiles; AFS = Agricultural feed stores; RSH = Residential settings; FAR = Farm/dairy/agricultural settings. Variable-importance rankings from random-forest classification of multistate, animal-contact NTS outbreaks. Panels A and B show the importance of exposure sources (mean decrease in Gini and permutation mean decrease in accuracy, respectively); panels C and D show the importance of exposure settings. Higher values indicate greater contribution to the model's ability to distinguish outbreak profiles. Birds and reptiles were the top source predictors, while agricultural feed stores and residential settings were the top setting predictors.

The community detection identified two dominant Louvain communities with non-zero network metrics (**Table 2**) and 29 smaller communities, each consisting of a single node, all of which had no network metrics (S4 Table).

Two main Louvain communities were identified. Community 1 consisted of nodes representing states, NTS serovars, exposure sources, and settings. It included the following states: Virginia, Pennsylvania, Ohio, Kentucky, Georgia, Indiana, Michigan, Tennessee, Illinois, Kansas, Massachusetts, Maryland, Maine, Vermont, Louisiana, Arkansas, Mississippi, Delaware, and Connecticut. The serovars in Community 1 were Johannesburg, Infantis, Enteritidis, Altona, Mbandaka, Muenster, and Agbeni. Exposure sources included birds, and the exposure settings included agricultural feed stores, dairy/farm/agricultural settings, educational settings (school/college/university), and child daycare/preschool.

Community 2 consisted of nodes representing states, serovars, exposure sources, and settings. The states included in community 2 were Washington, Texas, California, Oklahoma, Missouri, Wisconsin, Minnesota, Utah, Nebraska, Colorado, Nevada, Florida, Oregon, Arizona, Iowa, Idaho, Montana, and Wyoming. The serovars in community 2 were Muenchen, Hadar, Typhimurium, Braenderup, Berta, Pomona, Montevideo, Newport, Indiana, I 4,[5],12:i:-, Thompson, Sandiego, Poona, Uganda, and Cotham. Animal sources included reptiles, mammals, and amphibians. The exposure setting was residential settings.

Among the 48 contiguous U.S. states and the District of Columbia, eleven states, North Carolina, South Carolina, South Dakota, New York, New Hampshire, North Dakota, New Mexico, Rhode Island, West Virginia, and New Jersey, had

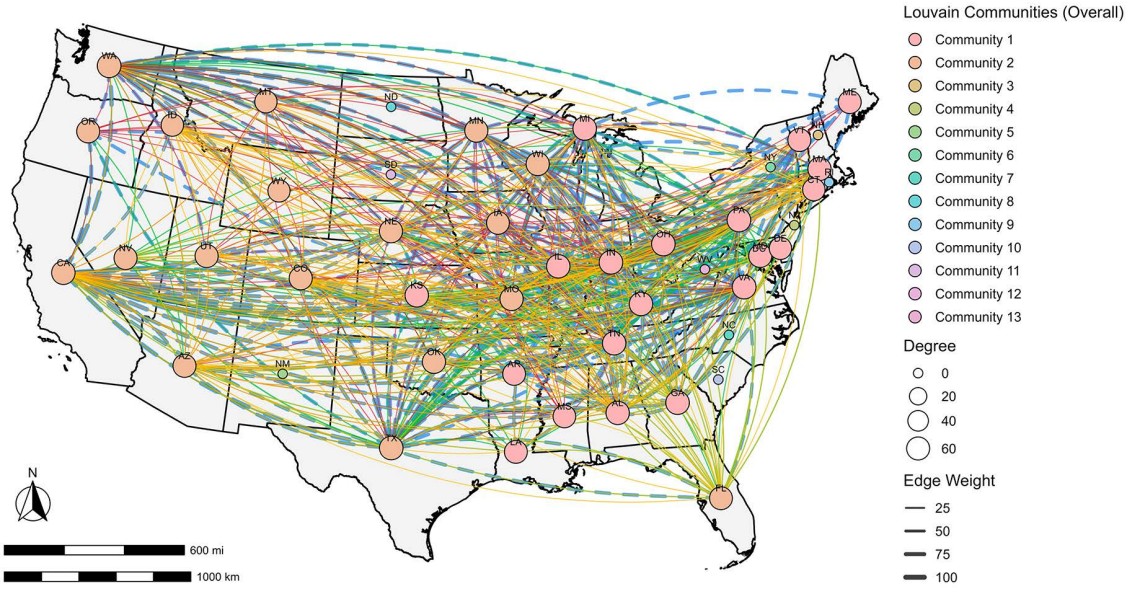

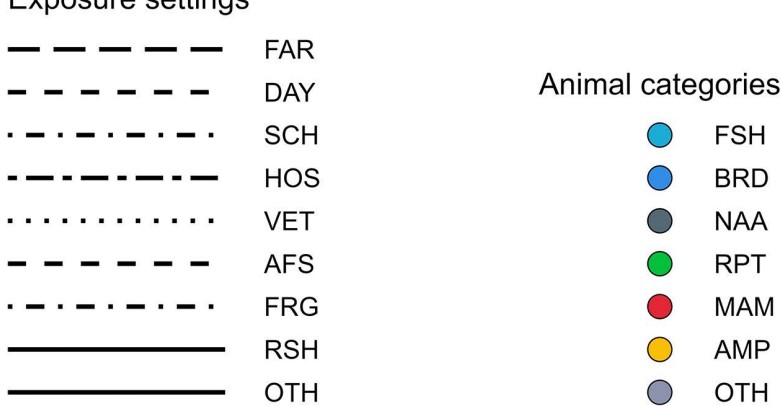

**Fig 4. State-level outbreak co-occurrence network of multistate nontyphoidal *Salmonella enterica* outbreaks in the United States.** States: AL-Alabama, AZ-Arizona, AR-Arkansas, CA-California, CO-Colorado, CT-Connecticut, DC-District of Colombia, DE-Delaware, FL-Florida, GA-Georgia, ID-Idaho, IL-Illinois, IN-Indiana, IA-Iowa, KS-Kansas, KY-Kentucky, LA-Louisiana, ME-Maine, MD-Maryland, MA-Massachusetts, MI-Michigan, MN-Minnesota, MS-Mississippi, MO-Missouri, MT-Montana, NE-Nebraska, NV-Nevada, NH-New Hampshire, NJ-New Jersey, NM-New Mexico, NY-New York, NC-North Carolina, ND-North Dakota, OH-Ohio, OK-Oklahoma, OR-Oregon, PA-Pennsylvania, RI-Rhode Island, SC-South Carolina, SD-South Dakota, TN-Tennessee, TX-Texas, UT-Utah, VT-Vermont, VA-Virginia, WA-Washington, WV-West Virginia, WI-Wisconsin, WY-Wyoming. Animal Sources: BRD-Birds, RPT-Reptiles, MAM-Mammals. Exposure settings: OTH-Others, AFS-Agricultural feed stores, RSH-Residential settings, FAR-Farm/dairy/agricultural settings, DAY-Child daycare/preschool, SCH-School/college/university, HOS-Hospital, VET-Veterinary clinic, FRG-Fairground.

zero connectivity (Degree = 0). Several exposure sources and settings also had zero connectivity (S4 Table). These variables were included in the Louvain community analysis, but their co-occurrence with other variables was too low to yield meaningful connections in the network. The data transformation process, where the presence-absence matrix (including

**Table 2. Network metrics for nodes in the multistate outbreak co-occurrence network of nontyphoidal *Salmonella enterica* animal-contact related outbreaks in the United States.**

| Node | Label | Type | Degree | Strength | Betweenness | Closeness | Clustering | Community |
|------|-------|------|--------|----------|-------------|-----------|------------|-----------|
| BRD | Birds | Source | 59 | 3733 | 11.08 | 0.33 | 0.77 | 1 |
| SJHB | Johannesburg | Serovar | 14 | 22 | 45.04 | 0.40 | 0.9 | 1 |
| SINF | Infantis | Serovar | 43 | 828 | 0.46 | 0.30 | 0.97 | 1 |
| SEN | Enteritidis | Serovar | 43 | 1019 | 0 | 0.24 | 0.98 | 1 |
| SALT | Altona | Serovar | 21 | 85 | 3.42 | 0.27 | 1 | 1 |
| SMBK | Mbandaka | Serovar | 38 | 171 | 9.89 | 0.38 | 1 | 1 |
| SMUE2 | Muenster | Serovar | 25 | 49 | 90.46 | 0.43 | 1 | 1 |
| SAGB | Agbeni | Serovar | 25 | 35 | 53.19 | 0.46 | 1 | 1 |
| AFS | Agricultural feed stores | Setting | 52 | 1237 | 25.30 | 0.32 | 0.89 | 1 |
| FAR | Dairy/farm/agricultural settings | Setting | 40 | 125 | 101.66 | 0.39 | 0.97 | 1 |
| SCH | School/college/university (educational settings) | Setting | 23 | 46 | 1.21 | 0.33 | 1 | 1 |
| DAY | Child daycares/preschools | Setting | 5 | 5 | 26.94 | 0.37 | 1 | 1 |
| VA | Virginia | State | 67 | 3465 | 20.29 | 0.42 | 0.72 | 1 |
| PA | Pennsylvania | State | 66 | 3085 | 10.90 | 0.43 | 0.74 | 1 |
| OH | Ohio | State | 65 | 3157 | 17.19 | 0.43 | 0.75 | 1 |
| KY | Kentucky | State | 64 | 2800 | 27.80 | 0.44 | 0.75 | 1 |
| GA | Georgia | State | 65 | 2689 | 61.76 | 0.46 | 0.75 | 1 |
| IN | Indiana | State | 65 | 2669 | 19.39 | 0.45 | 0.76 | 1 |
| MI | Michigan | State | 65 | 2703 | 41.00 | 0.45 | 0.76 | 1 |
| TN | Tennessee | State | 64 | 3174 | 10.27 | 0.42 | 0.77 | 1 |
| IL | Illinois | State | 64 | 3150 | 16.95 | 0.43 | 0.77 | 1 |
| KS | Kansas | State | 64 | 2265 | 75.45 | 0.47 | 0.77 | 1 |
| MA | Massachusetts | State | 62 | 2655 | 21.88 | 0.43 | 0.79 | 1 |
| MD | Maryland | State | 61 | 2421 | 14.55 | 0.42 | 0.8 | 1 |
| ME | Maine | State | 59 | 1996 | 29.02 | 0.40 | 0.81 | 1 |
| VT | Vermont | State | 60 | 2034 | 14.03 | 0.43 | 0.81 | 1 |
| LA | Louisiana | State | 59 | 2010 | 32.95 | 0.45 | 0.82 | 1 |
| AR | Arkansas | State | 58 | 2343 | 10.83 | 0.41 | 0.83 | 1 |
| MS | Mississippi | State | 57 | 1882 | 12.55 | 0.41 | 0.84 | 1 |
| DE | Delaware | State | 54 | 901 | 53.90 | 0.39 | 0.87 | 1 |
| CT | Connecticut | State | 55 | 1901 | 2.81 | 0.40 | 0.87 | 1 |
| RPT | Reptiles | Source | 53 | 480 | 71.72 | 0.41 | 0.84 | 2 |
| MAM | Mammals | Source | 37 | 96 | 146.51 | 0.42 | 0.98 | 2 |
| AMP | Amphibians | Source | 33 | 33 | 279.91 | 0.56 | 1 | 2 |
| SMUE | Muenchen | Serovar | 36 | 130 | 10.06 | 0.39 | 0.98 | 2 |
| SHDR | Hadar | Serovar | 42 | 551 | 0.14 | 0.29 | 0.99 | 2 |
| STY | Typhimurium | Serovar | 43 | 379 | 0 | 0.30 | 0.99 | 2 |
| SBRP | Braenderup | Serovar | 41 | 268 | 0.02 | 0.31 | 1 | 2 |
| SBRT | Berta | Serovar | 4 | 4 | 43.59 | 0.35 | 1 | 2 |
| SPMN | Pomona | Serovar | 32 | 99 | 17.67 | 0.39 | 1 | 2 |
| SMTV | Montevideo | Serovar | 35 | 122 | 23.50 | 0.39 | 1 | 2 |
| SNWP | Newport | Serovar | 36 | 100 | 24.20 | 0.40 | 1 | 2 |
| SIND | Indiana | Serovar | 30 | 115 | 35.48 | 0.40 | 1 | 2 |
| STM | I 4,[5],12:i:- | Serovar | 37 | 126 | 31.61 | 0.41 | 1 | 2 |
| STHM | Thompson | Serovar | 24 | 52 | 77.48 | 0.41 | 1 | 2 |

*(Continued)*

**Table 2.** (Continued)

| Node | Label | Type | Degree | Strength | Betweenness | Closeness | Clustering | Community |
|------|-------|------|--------|----------|-------------|-----------|------------|-----------|
| SSDG | Sandiego | Serovar | 21 | 34 | 78.26 | 0.42 | 1 | 2 |
| SPON | Poona | Serovar | 29 | 70 | 96.77 | 0.43 | 1 | 2 |
| SUGA | Uganda | Serovar | 22 | 22 | 161.01 | 0.48 | 1 | 2 |
| SCTH | Cotham | Serovar | 28 | 28 | 241.31 | 0.52 | 1 | 2 |
| STYC | Typhimurium var Copenhagen | Serovar | 33 | 33 | 279.91 | 0.56 | 1 | 2 |
| RSH | Residential settings | Setting | 51 | 326 | 53.62 | 0.42 | 0.91 | 2 |
| WA | Washington | State | 66 | 2864 | 74.73 | 0.45 | 0.74 | 2 |
| TX | Texas | State | 65 | 3101 | 19.38 | 0.44 | 0.76 | 2 |
| CA | California | State | 64 | 3301 | 8.58 | 0.41 | 0.77 | 2 |
| OK | Oklahoma | State | 62 | 1944 | 30.35 | 0.43 | 0.79 | 2 |
| MO | Missouri | State | 62 | 2822 | 4.84 | 0.42 | 0.8 | 2 |
| WI | Wisconsin | State | 62 | 2643 | 15.46 | 0.44 | 0.8 | 2 |
| MN | Minnesota | State | 61 | 2710 | 25.25 | 0.44 | 0.8 | 2 |
| UT | Utah | State | 61 | 2186 | 64.38 | 0.46 | 0.8 | 2 |
| NE | Nebraska | State | 61 | 2352 | 9.30 | 0.43 | 0.81 | 2 |
| CO | Colorado | State | 61 | 2331 | 19.92 | 0.44 | 0.81 | 2 |
| NV | Nevada | State | 60 | 1577 | 26.09 | 0.44 | 0.81 | 2 |
| FL | Florida | State | 59 | 2133 | 5.70 | 0.42 | 0.82 | 2 |
| OR | Oregon | State | 60 | 2163 | 23.77 | 0.44 | 0.82 | 2 |
| AZ | Arizona | State | 60 | 2207 | 38.55 | 0.44 | 0.82 | 2 |
| IA | Iowa | State | 58 | 2385 | 0.12 | 0.29 | 0.83 | 2 |
| ID | Idaho | State | 55 | 1724 | 42.22 | 0.44 | 0.88 | 2 |
| MT | Montana | State | 54 | 1645 | 34.59 | 0.43 | 0.89 | 2 |
| WY | Wyoming | State | 53 | 1440 | 23.96 | 0.39 | 0.92 | 2 |

U.S. states, exposure sources, and exposure settings) was converted into a co-occurrence matrix, retained only variables with co-occurrence frequencies greater than 5%. As a result, these variables were included in the analysis but had zero connectivity, reflecting minimal involvement in shared outbreaks with other states or exposure categories. (S4 Table).

A total of 38 states exhibited non-zero connectivity, and their degree values ranged from 53 to 67 across the connected states, serovars, exposure sources, and settings. Among these states, Virginia had the highest degree value of 67, followed by Pennsylvania and Washington (66 each). Among serovars, the highest degree of 43 was observed for Typhimurium, Enteritidis, and Infantis. Among exposure sources, birds showed the highest degree value of 59, followed by reptiles (53). Among exposure settings, agricultural feed stores had the highest degree value of 52, followed by residential settings (40), and farm/dairy/agricultural settings.

The strength ranged between 901 and 3465. Among states, Virginia had the highest strength value of 3465, followed by California (3301), Tennessee (3174), and Ohio (3157). Among exposure sources, birds showed the highest strength value of 3733, followed by reptiles (480) and mammals (96). Among serovars, the highest strength was observed for Enteritidis (1019), Infantis (828), Hadar (551), and Typhimurium (379). Among exposure sources, the highest strength was observed for agricultural feed stores (1237) and residential settings (326) (**Table 2**).

The betweenness centrality ranged between 0.02 and 279.91. Among states, Kansas (75.46), Washington (74.72), Utah (64.38), Georgia (61.75), and Delaware (53.90) exhibited the highest betweenness. Among exposure sources, amphibians showed the highest betweenness (279.91), followed by mammals (146.51). Among serovars, the highest

betweenness was observed for Typhimurium var Copenhagen (279.91) and Cotham (241.30). Among exposure sources, the highest betweenness was for farm/dairy/agricultural settings (101.67).

The closeness centrality ranged between 0.24 and 0.56. Among states, the highest closeness was observed for Kansas (0.47), followed by Utah and Georgia (0.46 each). Among exposure sources, amphibians showed the highest closeness value (0.56), followed by mammals (0.42). Among serovars, Typhimurium var Copenhagen and Cotham (0.56 each) had the highest closeness value. While Enteritidis, Typhimurium, Braenderup, and Infantis had the lowest values. Among exposure sources, residential settings (0.42) and farm/dairy/agricultural settings (0.39) had high closeness values.

The clustering coefficients ranged between 0.72 and 1.00. Among states, Wyoming (0.92) and Montana (0.89) had the highest values. Among exposure sources, amphibians showed the highest clustering value of 1.00, followed by mammals (0.98). Among serovars, the highest clustering was observed for several, including Typhimurium var Copenhagen, Cotham, and Uganda. Among exposure settings, daycares and educational institutions had the highest values of 1.00, respectively (Table 2).

## Discussion

This study provides a national characterization of animal-contact-associated multistate NTS outbreaks in the U.S. over more than a decade of surveillance. An integrated approach was followed, which included the assessment of multistate NTS outbreak IR trends, exposure sources, settings, and interstate co-occurrence patterns, identifying distinct serovar-exposure source-exposure setting-region clusters that reflect distinct transmission pathways and could inform prevention strategies. Among exposure sources, birds and reptiles, among exposure settings, agricultural feed stores and residential homes, among serovars Enteritidis, Infantis, Hadar, Typhimurium, and Braenderup, and among regions, the Midwest, Northeast, and Southern regions contributed the most to NTS animal-contact related multistate outbreaks. These findings add to previous work on single-state NTS outbreak patterns [50,51], which were smaller, geographically localized, and most often associated with exposures to mammals in farm settings or homes, reflecting localized animal husbandry practices, facility-specific deficiencies, or household-level risk factors. In contrast, this current analysis on multistate NTS outbreaks demonstrated that multistate outbreaks involved interstate connectivity and were predominantly linked to birds and reptiles. Our findings align with multiple investigations from the U.S. that described the interstate movement through commercial distribution channels of infected live poultry via mail-order hatcheries [21,52] and reptiles and amphibians via trade networks and illegal sale/distribution of small turtles [53] can disseminate *Salmonella* spp. contaminated animals across wide geographic areas before human illness is detected. To prevent and control NTS multistate outbreaks, cross-jurisdictional surveillance, harmonized exposure assessments, educating at-risk populations (e.g., children) on non-food transmission routes of NTS, and prevention and control strategies that account for the national distribution systems of live birds and reptiles are suggested [52,53].

Findings from our study support strengthening upstream control of multistate NTS spread locations, including improving biosecurity measures at hatcheries, agricultural feed stores, reptile breeding facilities, and poultry-livestock shipment services [11,52]. The public health authorities should prioritize cross-jurisdiction collaborations and messaging during seasonal at-risk periods, and should incorporate standardized traceback investigations focusing on reptiles and poultry to reduce delays in identifying sources of transmission and spread [13,53].

The Joinpoint regression approach was used to examine temporal shifts in NTS multistate outbreak trends, which identified two distinct phases in multistate animal contact NTS outbreak incidence. The first period, from 2009 to 2013, showed a steep, statistically significant decline in IRs, and a second period from 2013 to 2022, where there was a non-significant but modest downward trend. Several factors may contribute to these patterns. The first period decrease might suggest upstream industry practices and regulatory actions that improved hatchery sanitation, expanded traceback capacity for live poultry and reptile distribution chains, and improved sequencing and subtyping of NTS isolates, aiding traceback investigations to link illnesses to exposure sources [21,52,53]. The second moderate decreasing phase might

be explained by the COVID-19 outbreak, when enteric disease investigations and reporting were impacted by resource allocations to contain the pandemic [54]. However, these findings should be interpreted with caution because our study evaluated national-level NTS outbreak trends, which might mask state-level local trends.

The data-driven supervised machine learning approach (Random Forest) identified important predictors, including animal contact exposure sources, settings, and States. Birds and reptiles, and outbreaks linked to agricultural feed stores and residential homes, were both the most commonly reported and the most informative variables in influencing the outcome of the analysis, suggesting that these sources and settings are frequent contributors to multistate animal-contact-related NTS outbreaks in the Northeast and Midwest and exhibit consistent patterns useful for outbreak classification and public-health targeting.

The co-occurrence network analysis revealed regional clustering and identified heterogeneity among animal contact sources and exposure settings at the human-animal-environmental interfaces across the U.S. regions in the Midwest, Northeast, and South were high-burden states for poultry-associated outbreaks and were linked to exposure settings of agricultural feed stores, private homes, and serovars Infantis, Enteritidis, Altona, and Muenster, which have been described previously as a common source and setting for backyard poultry associated human infections [21,52].

The second network cluster included states in the West, Mountain West, and Southern U.S., with mammal, reptile, and amphibian exposure sources in residential settings. The serovars linked to this network included Typhimurium, I 4, [5],12:i:-, Muenchen, Braenderup, Hadar, Berta, Pomona, Montevideo, Newport, Indiana, Thompson, Poona, and Sandiego, Cotham, and Uganda, serovars more commonly linked to reptiles and amphibians [55]. The findings of our study are consistent with previous studies showing reptile and amphibian-associated NTS transmission in the West and South, where exotic pet ownership and wildlife were major risks of NTS transmission [53].

Serovar Typhimurium showed a broad host range, co-occurred with different exposure settings, in a wide multistate network, suggesting a varied adaptability, posing an increased risk. This common serovar, with overlapping exposure sources and transmission patterns, requires cross-sectoral interventions.

Prioritization of regions based on common reservoirs, serovars, and exposure settings should be considered when developing prevention and control strategies. In regions with high exotic pet adoptions/populations (reptiles/small mammalian household pets), education and outreach campaigns should focus on infection control practices when handling reptiles and exotic pets, obtaining reptiles from trusted breeders, and avoiding high-risk species [15,53,56]

## Limitations

A key limitation of our co-occurrence analytical approach was the lack of information regarding the state of origin for multistate outbreaks. Additionally, the NORS outbreak data are subject to underreporting due to differences in reporting capacity across jurisdictions and a lack of uniformity in reported exposure sources or settings. Also, not all individuals with gastrointestinal illness seek medical care due to barriers such as limited access to healthcare, lack of insurance, socio-economic constraints, or reluctance to visit a clinic for self-limiting symptoms. In addition, NORS is a passive, voluntary surveillance platform, and reporting completeness might vary across states and over time. Finally, the disease burden presented reflects data available through August 2024, when the dataset was made available for this study, while ongoing investigations and delayed reporting may influence future estimates [13], and interpretation of our results should be made with caution. Future studies should evaluate the most recent NTS outbreak trends.

## Conclusion

This study examined animal contact-related multistate NTS outbreaks across the U.S. between 2009 and 2022. The study findings suggest that outbreaks in different regions (communities) were associated with distinct serovar profiles, exposure sources, and settings, highlighting potential geographical and environmental factors that drive the spread of NTS infections. Machine learning and network analysis identified key serovars (Enteritidis, Typhimurium, Infantis, Hadar) and

exposure sources (birds, reptiles, mammals) and settings (agricultural feed stores, residential, and farms), in the Midwest, Northeast, and Mountain West, as influential factors shaping outbreak patterns. Temporal trends in the NTS outbreak IRs indicated a stabilizing pattern in later years. These findings highlight the complexity of NTS transmission dynamics across human, animal, and environmental interfaces, which traditional epidemiological methods cannot fully capture. Integrating advanced analytical tools into disease surveillance can enhance understanding of the factors driving NTS persistence and transmission. Future research should refine these models to support public health initiatives aimed at reducing NTS disease burden.

## Public health implications

Our study supports the need for targeted interventions at key multi-state NTS outbreak drivers, including poultry and livestock operations, live poultry and reptile distribution supply chains, mail-order hatcheries, agricultural feed stores, and residential settings. The presence of distinct regional clusters asks for region-specific prevention strategies rather than uniform national approaches. Strengthening upstream controls, enhancing biosecurity, and improving hygiene practices in high-risk settings can help disrupt interstate transmission. Finally, standardized and timely reporting, modernization of surveillance systems, and improved communication between federal agencies and state and local health departments could enhance early detection of multistate outbreaks and increase the overall effectiveness of pathogen-reduction efforts.

## Supporting information

**S1 Table. National-level nontyphoidal Salmonella enterica animal contact-related multistate outbreak and illness incidence rates across the U.S. 2009–2022.** The S1 Table represents the national NTS multistate outbreak incidence rates (Per 10 Million Population-Years).
(DOCX)

**S2 Table. Trend analysis results of the Joinpoint regression models of nontyphoidal *Salmonella enterica* multistate outbreak incidence rates at the national level across the U.S., 2009–2022.** The output from which the appropriate model was selected for the graph by the software based on BIC and AIC values. APC (Annual Percent Change), Segment Start (Point where Joinpoint is inserted), Segment end (where next Joinpoint is inserted or trend stabilized), APC 95% LCL (Annual Percent Change 95% Lower Confidence Limit), APC 95% UCL (APC 95% Upper Confidence Limit). The Joinpoint regression analysis presents different models with joinpoints, significant and nonsignificant temporal shifts over time.
(DOCX)

**S3 Table. Random Forest classifier ranking (relative importance) of categorical variables in nontyphoidal *Salmonella enterica* animal contact-related multistate outbreaks across the US., 2009–2022.** The above table indicates the Random Forest (RF) classifier ranking of categorical variables. Each category's rank (relative importance) has been mentioned.
(DOCX)

**S4 Table. Nontyphoidal *Salmonella enterica* serovars animal contact-related multistate outbreaks co-occurrence network and Louvain community memberships of nodes with no node metrics across the U.S. 2009–2022.** The co-occurrence network analysis of NTS multistate outbreaks presents the US states, NTS serovars, animal exposure sources, and settings as nodes. Degree is the number of connections each node has, strength is the frequency of shared outbreaks, Betweenness: the number of times a node is on the shortest path between two nodes, Closeness: how close the node is to all other nodes in the network, Clustering Coefficient: a measurement of local connectivity, and Louvain Communities: a group of densely connected nodes. States: NH-New Hampshire, NJ-New Jersey,

NM-New Mexico, NY-New York, NC-North Carolina, ND-North Dakota, RI-Rhode Island, SC-South Carolina, SD-South Dakota, WV-West Virginia, DC-District of Columbia. Animal Sources: FSH-Fish, Exposure settings; OTH-Others, HOS-Hospital, VET-Veterinary clinic, FRG-Fairground, Serovars; Serovars: SPBT-Paratyphi B var. L(+) tartrate +, STP-Saintpaul, SJAV-Javiana,STLK-Telelkebir, UNK-unknown, SHDL-Heidelberg, SHFD-Hartford, SDRB-Durban, SGMN-Gaminara, SLML-Lomalinda, SSTN-Stanley, SHVN-Havana, SLIT-Litchfield, SAPP-Apapa, SFLU-Fluntern, SAGO-Agona, SOFF-Offa, SOTH-other, SALA-Alachua, SANA-Anatum, SMAN-Manhattan, SORA-Oranienburg, SGUN-Guinea, SLIV-Liverpool, SEAL-Ealing, SBAN-Banana, SVIT-Vitkin, SLOM-Lome, SSB3-Subspecies IIIb. (DOCX)

## Acknowledgments

We want to acknowledge the Centers for Disease Control and Prevention (CDC) and the National Outbreak Reporting System (NORS) for providing access to the dataset.

The findings and conclusions in this report are those of the authors and do not necessarily represent the official position of the Centers for Disease Control and Prevention.

## Author contributions

**Conceptualization:** Suman Bhowmick, Csaba Varga.

**Data curation:** Hammad Ur Rehman Bajwa.

**Formal analysis:** Hammad Ur Rehman Bajwa, Suman Bhowmick.

**Investigation:** Hammad Ur Rehman Bajwa, Csaba Varga.

**Methodology:** Hammad Ur Rehman Bajwa, Suman Bhowmick, Csaba Varga.

**Project administration:** Csaba Varga.

**Resources:** Csaba Varga.

**Supervision:** Csaba Varga.

**Validation:** Hammad Ur Rehman Bajwa.

**Visualization:** Hammad Ur Rehman Bajwa.

**Writing – original draft:** Hammad Ur Rehman Bajwa, Csaba Varga.

**Writing – review & editing:** Hammad Ur Rehman Bajwa, Suman Bhowmick, Csaba Varga.

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
