## [Decision Letter · Decision Letter 0]

20 Apr 2026

PONE-D-26-09918Multistate Animal-Contact-Related Nontyphoidal Salmonella enterica Outbreaks in the United States, 2009-2022: Network and Machine Learning Analyses of Exposure Sources, Settings, and SerovarsPLOS One

Dear Dr. Varga,

Thank you for submitting your manuscript to PLOS ONE. After careful consideration, we feel that it has merit but does not fully meet PLOS ONE’s publication criteria as it currently stands. Therefore, we invite you to submit a revised version of the manuscript that addresses the points raised during the review process.

Your manuscript has been reviewed by an expert in your field and a major revision is needed before a decision can be made..

We look forward to receiving your revised manuscript.

Kind regards,

Yung-Fu Chang

Academic Editor

PLOS One

4. We note that Figure 4 in your submission contain [map/satellite] images which may be copyrighted. All PLOS content is published under the Creative Commons Attribution License (CC BY 4.0), which means that the manuscript, images, and Supporting Information files will be freely available online, and any third party is permitted to access, download, copy, distribute, and use these materials in any way, even commercially, with proper attribution. For these reasons, we cannot publish previously copyrighted maps or satellite images created using proprietary data, such as Google software (Google Maps, Street View, and Earth). For more information, see our copyright guidelines: http://journals.plos.org/plosone/s/licenses-and-copyright.

1. You may seek permission from the original copyright holder of Figure 4 to publish the content specifically under the CC BY 4.0 license.

5. In the online submission form, you indicated that [Data cannot be shared publicly because of data protection. Data are available from the CDC NORS via data request.].

Reviewers' comments:

Reviewer's Responses to Questions

**Comments to the Author**

1. Is the manuscript technically sound, and do the data support the conclusions?

Reviewer #1: Yes

2. Has the statistical analysis been performed appropriately and rigorously? 

Reviewer #1: Yes

3. Have the authors made all data underlying the findings in their manuscript fully available?

Reviewer #1: No

4. Is the manuscript presented in an intelligible fashion and written in standard English?

Reviewer #1: Yes

5. Review Comments to the Author

Reviewer #1: Overall comment: The authors conducted a study titled “Multistate Animal-Contact-Related Nontyphoidal Salmonella enterica Outbreaks in the United States, 2009-2022: Network and Machine Learning Analyses of Exposure Sources, Settings, and Serovars” that analyzed animal contact-related non-typhoidal Salmonella (NTS) outbreak data, which was utilized for calculating incidence rate per 10 million population‑years, assessed temporal trends using Join point regression, co‑occurrence network linking, and machine learning classification algorithm, such as a random forest classifier to identify variables most useful for distinguishing outbreak profiles. Overall, the manuscript is well written and provides important information on the burden of multistate NTS outbreaks associated with animal contact. This manuscript addresses an important one-health issue. However, the following comments need to be addressed to improve the manuscript clarity and readability.

Specific comment:

Introduction:

Comment: The author should include a brief introduction about the public health burden of non-typhoidal Salmonella (NTS), including the predominant NTS serovars in the United States.

Comment: The relevance of the study is unclear. The author should clarify why this study findings are important (by addressing the “so what?” question in the introduction section).

Materials and Methods:

Comment: The author should provide a brief description of how animal contact related NTS outbreak surveillance data were collected to help the reader understand the data source, collection, population, and reporting process.

Comment: The author should describe how the surveillance data is assessed, including missing values, value distributions, and duplication, before preparing the analytic dataset for this study.

Comment: Authors should provide a clear case definition of 'animal contact related NTS outbreak' for better understanding by the readers of the paper. Additionally, the author should mention the inclusion criteria for including ‘animal contact related NTS outbreak' events for analysis in this study.

Comment: The author should report the study design in the methods section.

Comment (Line 116): The author should provide a full elaboration of ‘NORS’ before introducing the abbreviation.

Comment (Line 145): Authors should provide a table that displays all explanatory variables (features), along with their categories and types, to enhance reader understanding about the variables of interest in this study.

Comment: The author should provide a logical order of data analysis, with their headings, to test each hypothesis in this study

Comment: The author should specify the version of the Joinpoint Regression (JPR) software used in this study.

Comment (Line 144): What is the reasoning behind exclusively using the random forest model (RF)? How do the authors partition (%) the data into training and testing sets for the RF model? The author should provide this information. Did the author try any alternative classification algorithm (e.g., logistic regression ….) as part of the sensitivity analysis? What form of cross‑validation was used to assess the RF model's performance? The author should specify the R packages used for the RF modeling.

Comment: The author should provide a brief introduction and a rationale for the network analysis and explain why the network is treated as undirected and weighted. Additionally, the author should specify the R packages used in the study's network analysis.

Comment (Line 144): In the RF model section, the author should specify the outcome variable and explain how it is defined. Also, specify the predictor of interest and explain how the predictors were chosen for inclusion in the model.

Comment: It would be helpful if the author included the full R scripts for the RF model and network analyses as supplementary files to help readers and learners understand the analyses.

Results:

Comment (Line 190-192): These sentences should be in the method section, not part of the result section of the manuscript.

Comment (Line 193-194): These sentences should be placed in the discussion section, not in the results section. In the RF model results, the reader can expect to see the RF model performance matrices and the variable importance identified in the RF model (top, second, and third variables) as the strongest predictors of the outcome variable of interest.

Comment (Line 228-232): These sentences are not results; they are part of the discussion. The author should remove this section from the results section and place it in the appropriate section in the discussion.

Discussion

Comment: The author should provide a primary conclusion statement that logically and defensibly draws from the study results.

6. PLOS authors have the option to publish the peer review history of their article (what does this mean?). If published, this will include your full peer review and any attached files.

Reviewer #1: **Yes:** Shamim Sarkar

---

## [Author Response · Author response to Decision Letter 1]

5 May 2026

We are thankful to the editor and reviewers for their constructive feedback. We have addressed all comments to improve the organization, clarity, and transparency of our manuscript. The revisions have been incorporated into the updated manuscript, as described below.

Response to Journal Requirements

Comment

Response:

The manuscript follows PLOS ONE's style requirements as outlined in the provided templates. Please see the revised manuscript file.

Response:

The codes used in the analysis will be shared upon acceptance of our work.

Comment

Response:

The in-text citations were updated according to PLOS ONE requirements. The supporting information file has also been updated accordingly.

4. We note that Figure 4 in your submission contain [map/satellite] images which may be copyrighted. All PLOS content is published under the Creative Commons Attribution License (CC BY 4.0), which means that the manuscript, images, and Supporting Information files will be freely available online, and any third party is permitted to access, download, copy, distribute, and use these materials in any way, even commercially, with proper attribution. For these reasons, we cannot publish previously copyrighted maps or satellite images created using proprietary data, such as Google software (Google Maps, Street View, and Earth). For more information, see our copyright guidelines: http://journals.plos.org/plosone/s/licenses-and-copyright.

1. You may seek permission from the original copyright holder of Figure 4 to publish the content specifically under the CC BY 4.0 license.

Response:

The base maps were generated using TIGER/Line shapefiles provided by the United States Census Bureau. These are in the public domain and not subject to copyright. Under the US copyright act (USC § 105), works of the US government are not eligible for copyright protection. Therefore, the maps are freely available for public use and comply with the CC BY 4.0 licensing requirements. Additionally, we have indicated this in our data source section of materials and methods.

Response:

Not applicable to our situation. Please see our response above.

5. In the online submission form, you indicated that [Data cannot be shared publicly because of data protection. Data are available from the CDC NORS via data request.].

Response:

We updated the data availability statement as follows: “The data underlying the findings described in the manuscript are available within the manuscript and in the supplementary files.“

Response to Reviewers

The author should include a brief introduction about the public health burden of non-typhoidal Salmonella (NTS), including the predominant NTS serovars in the United States.

Response:

Thank you for your suggestion. We included a section on the disease burden and main NTS serovars:

“Globally, NTS causes over 95 million enteric illnesses, with more than 500,000 invasive infections, leading to high fatality rates [2]. It also poses an economic burden, contributing to millions of disability-adjusted life years and to substantial healthcare costs worldwide [3]. Most of NTS infections are self-limiting; however, severe invasive disease can occur, primarily affecting immunocompromised individuals, infants, children, and the elderly, often resulting in death [4]. The NTS bacteria are adaptable to various hosts, with NTS serovars such as Typhimurium, Enteritidis, Heidelberg, and Infantis being common causes of human enteric disease, including in the U.S. [5]. “

Comment:

The relevance of the study is unclear. The author should clarify why this study findings are important (by addressing the “so what?” question in the introduction section).

Response:

We updated the introduction section to address your comment and included the following:

“No previous study explored long-term, NTS serovar-level animal contact-related multistate outbreaks in the US, and the relationship between serovars and animal contact sources remains unclear. Previous epidemiological studies focused on individual serovars or specific exposure sources, limiting the full scope of NTS transmission dynamics across the U.S. [16,21,22]. In multistate outbreaks, interconnected events might involve shared distribution systems, supply chains, and occupational exposures [21], and an integrated analytical approach is needed to quantify NTS burden and identify factors that sustain its endemicity across various reservoirs and regions.”

Comment:

The author should provide a brief description of how animal contact related NTS outbreak surveillance data were collected to help the reader understand the data source, collection, population, and reporting process.

Response:

Thank you for your suggestions.

WE included a section on NORS and included the following:

“Data source, study design, and setting

The National Outbreak Reporting System (NORS) is a voluntary disease surveillance system managed by the Centers for Disease Control and Prevention (CDC) that collects reports on enteric disease outbreaks, including those transmitted through animal contact. Animal contact is defined as any interaction between humans and animals that may result in the transmission of infectious agents. This includes direct contact with animals, such as handling, feeding, or caring for them, and indirect contact through exposure to animal environments, such as barns, petting zoos, or farms. Exposure sources include livestock, poultry, reptiles, and other animals, and outbreaks are included in this category where animal contact is suspected or confirmed as a contributing factor to the transmission of infectious diseases. [23, 24]. The NORS is used by local health departments, state-level health authorities, and the CDC to report and track outbreaks [24]. An outbreak is defined as the occurrence of more cases than expected within a period, a specific location, and a target population [25]. Multistate (more than 2 states) NTS outbreak investigations are coordinated and reported to NORS by CDC staff, while single-state outbreaks are investigated and reported to NORS by single jurisdictions [26].”

Comment:

The author should describe how the surveillance data is assessed, including missing values, value distributions, and duplication, before preparing the analytic dataset for this study.

Response:

We included all the available data in NORS. To explain our study approach, we included the following: This retrospective observational study analyzes human NTS animal contact-related multistate outbreaks. It includes all outbreaks with confirmed or suspected status, along with data on serovars, exposure states, animal contact exposure sources, and settings. Outbreaks with “not available” or "other" categories for serovars, exposure sources, and settings were treated as distinct categories for further analysis.”

Comment:

Authors should provide a clear case definition of 'animal contact related NTS outbreak' for better understanding by the readers of the paper. Additionally, the author should mention the inclusion criteria for including ‘animal contact related NTS outbreak' events for analysis in this study.

Response:

Thank you for your comment. We included the definition of “animal-contact outbreaks, and included the following:

“Animal contact is defined as any interaction between humans and animals that may result in the transmission of infectious agents. This includes direct contact with animals, such as handling, feeding, or caring for them, and indirect contact through exposure to animal environments, such as barns, petting zoos, or farms. Exposure sources include livestock, poultry, reptiles, and other animals, and outbreaks are included in this category where animal contact is suspected or confirmed as a contributing factor to the transmission of infectious diseases.”

Comment:

The author should report the study design in the methods section.

Response:

We included the following: “This retrospective observational study analyzes human NTS animal contact-related multistate outbreaks.”

Comment:

(Line 116): The author should provide a full elaboration of ‘NORS’ before introducing the abbreviation.

Response:

We corrected this and defined NORS before the abbreviation.

Comment:

(Line 145): Authors should provide a table that displays all explanatory variables (features), along with their categories and types, to enhance reader understanding about the variables of interest in this study.

Response:

Thank you for your suggestion. We included a Table at the end of the Descriptive Statistics section to describe the variables used in our analysis.

We included the following: “Table 1 summarizes the explanatory variables (features) used in the analysis, including their description, type, category, and the specific analysis in which they were applied.

Table 1. Summary of variables used for the analysis of Nontyphoidal Salmonella enterica animal contact-related multistate outbreaks across the US, 2009 – 2022“

Comment:

The author should provide a logical order of data analysis, with their headings, to test each hypothesis in this study

Response:

We provided a logical data analysis starting from general and more complex analyses. We included subsection titles for each step, as follows:

Assessing NTS outbreak incidence rates

Temporal trend analysis using the Joinpoint regression method

Random Forest analysis of multi-state outbreak data

Network analysis of multi-state outbreak data

Comment:

The author should specify the version of the Joinpoint Regression (JPR) software used in this study.

Response:

The version of the Joinpoint Regression software was added.

Comment

(Line 144): What is the reasoning behind exclusively using the random forest model (RF)?

Response

We have used the RF model as it can handle sparse reported binary surveillance data and to learn the nonlinear relationships amongst the different feature variables embedded into the model. We updated the manuscript, and included the following: “We analyzed the binary dataset of human NTS multistate outbreaks from 2009 to 2022 using the Random Forest (RF) modeling approach. The RF method is a flexible, non-parametric ensemble learning technique that can perform well with high-dimensional datasets, which contain correlated and binary predictors, as our surveillance dataset carries the same characteristics. The RF method can also model complex, non-linear relationships among the higher-dimensional model variables, and it can also include higher-order interactions without requiring explicit specifications. In addition, RF also incorporates internally validated performance estimates through out-of-bag (OOB) error and can generate interpretable variable importance measures, which were central to our objective of identifying the key driving factors distinguishing cluster membership into the model [36]. The RF model is less sensitive to multicollinearity and overfitting than traditional regression-based approaches, and is suitable for exploratory analyses of complex multistate outbreak data with many potentially correlated indicators.”

Comment

How do the authors partition (%) the data into training and testing sets for the RF model?

Response

We didn’t implement an explicit training/test split or cross-validation procedure. We have checked the model performance through the out-of-bag (OOB) error estimation inherent to the random forest algorithm; this has now been clarified in the manuscript.

We included the following: “In addition, RF also incorporates internally validated pe

---

## [Decision Letter · Decision Letter 1]

21 May 2026

Multistate animal-contact-related nontyphoidal Salmonella enterica outbreaks in the United States, 2009-2022: Network and machine learning analyses of exposure sources, settings, and serovars

PONE-D-26-09918R1

Dear Dr. Varga,

We’re pleased to inform you that your manuscript has been judged scientifically suitable for publication and will be formally accepted for publication once it meets all outstanding technical requirements.

Kind regards,

Yung-Fu Chang

Academic Editor

PLOS One

Additional Editor Comments (optional):

Reviewers' comments:

Reviewer's Responses to Questions

**Comments to the Author**

1. If the authors have adequately addressed your comments raised in a previous round of review and you feel that this manuscript is now acceptable for publication, you may indicate that here to bypass the “Comments to the Author” section, enter your conflict of interest statement in the “Confidential to Editor” section, and submit your "Accept" recommendation.

Reviewer #1: All comments have been addressed

2. Is the manuscript technically sound, and do the data support the conclusions?

Reviewer #1: Yes

3. Has the statistical analysis been performed appropriately and rigorously? 

Reviewer #1: Yes

4. Have the authors made all data underlying the findings in their manuscript fully available?

Reviewer #1: Yes

5. Is the manuscript presented in an intelligible fashion and written in standard English?

Reviewer #1: Yes

6. Review Comments to the Author

Reviewer #1: (No Response)

7. PLOS authors have the option to publish the peer review history of their article (what does this mean?). If published, this will include your full peer review and any attached files.

Reviewer #1: **Yes:** Shamim Sarkar

---

## [Editor Report · Acceptance letter]

PONE-D-26-09918R1

PLOS One

Dear Dr. Varga,

I'm pleased to inform you that your manuscript has been deemed suitable for publication in PLOS One. Congratulations! Your manuscript is now being handed over to our production team.

Kind regards,

on behalf of

Dr. Yung-Fu Chang

Academic Editor

PLOS One